# Automated Point Cloud Registration Approach Optimized for a Stop-and-Go Scanning System

**DOI:** 10.3390/s24010138

**Published:** 2023-12-26

**Authors:** Sangyoon Park, Sungha Ju, Minh Hieu Nguyen, Sanghyun Yoon, Joon Heo

**Affiliations:** Department of Civil and Environmental Engineering, Yonsei University, Seoul 03722, Republic of Korea; parksangyoon@yonsei.ac.kr (S.P.); jsh4907@yonsei.ac.kr (S.J.); hieuintelvn@gmail.com (M.H.N.); yoonssa@yonsei.ac.kr (S.Y.)

**Keywords:** point cloud registration, stop-and-go scanning systems, terrestrial laser scanning

## Abstract

The latest advances in mobile platforms, such as robots, have enabled the automatic acquisition of full coverage point cloud data from large areas with terrestrial laser scanning. Despite this progress, the crucial post-processing step of registration, which aligns raw point cloud data from separate local coordinate systems into a unified coordinate system, still relies on manual intervention. To address this practical issue, this study presents an automated point cloud registration approach optimized for a stop-and-go scanning system based on a quadruped walking robot. The proposed approach comprises three main phases: perpendicular constrained wall-plane extraction; coarse registration with plane matching using point-to-point displacement calculation; and fine registration with horizontality constrained iterative closest point (ICP). Experimental results indicate that the proposed method successfully achieved automated registration with an accuracy of 0.044 m and a successful scan rate (SSR) of 100% within a time frame of 424.2 s with 18 sets of scan data acquired from the stop-and-go scanning system in a real-world indoor environment. Furthermore, it surpasses conventional approaches, ensuring reliable registration for point cloud pairs with low overlap in specific indoor environmental conditions.

## 1. Introduction

### 1.1. Background

The digital transformation (DX) of construction sites has become increasingly important for achieving productivity growth. While DX for productivity improvement has been adopted across various industrial sectors since 2005, the construction industry has lagged behind, exhibiting the lowest digitization index and annual growth rate [1]. Recognizing this challenge, many leading construction companies have initiated the implementation of DX at their construction sites.

A key factor in successfully implementing these digital solutions is the effective management of data from facilities, which begins with acquiring accurate 3D geometric information. Point clouds, obtained through state-of-the-art field data acquisition systems and LiDAR scanners, have emerged as the prevailing approach for documenting the 3D geometry of existing buildings [2,3]. This method has gained popularity due to its ability to provide precise and comprehensive representations of structures, facilitating the overall DX of the construction sector.

Recent advancements in LiDAR scanning technology have revolutionized data acquisition by offering long-distance, high-resolution, and rapid scanning capabilities with millimeter-level accuracy [4,5]. Leveraging these advantages, laser scanning has found widespread applications across diverse fields, encompassing 3D model reconstruction [6,7,8], geometry quality control [9,10], construction assessment [11,12], historic preservation [13,14], and facility management [15].

The current laser scanning solutions for acquiring point cloud data can be broadly classified into two categories: (1) terrestrial laser scanning (TLS) and (2) mobile laser scanning (MLS). Each solution has its own set of advantages and limitations. TLS offers high-density point cloud data with exceptional accuracy, making it an ideal choice for generating reliable 3D geometric information. However, it collects data under static conditions, resulting in longer acquisition times compared to MLS [16]. On the other hand, MLS provides instant and continuous point cloud data, allowing for faster data collection. However, it comes with some drawbacks, such as lower point cloud resolution and higher noise due to motion distortion [17]. As a result, MLS may not be as suitable for generating precise 3D geometric information as TLS [18].

Progress in mobile robotics platforms has led to a revolutionary transformation in the process of acquiring high-resolution point clouds. These advancements have effectively addressed the drawbacks associated with TLS by significantly reducing the time required for scanning, resulting in a more efficient and time-effective process [19]. Many researchers have proposed stop-and-go scanning systems that incorporate mobile robots to collect precise 3D geometric data, thereby minimizing scanning time and human intervention [16,20,21,22,23,24]. In such systems, mobile robots navigate automatically to specific positions, where they conduct static laser scanning to generate detailed 3D point clouds.

However, the 3D point cloud collected by stop-and-go scanning systems requires essential post-processing known as registration, which involves aligning multiple point cloud datasets from local coordinate systems into a unified coordinate system that covers the entire scene [25,26]. While existing stop-and-go scanning systems can perform the acquisition of 3D point cloud data automatically, many of them do not address the issue of automatic point cloud registration or still rely on manual registration methods to align the acquired data [16,20], which can be highly inefficient and time-consuming.

### 1.2. Related Studies

#### 1.2.1. Point Cloud Registration

Previous research on TLS point cloud data registration has been conducted in various fields, including remote sensing, photogrammetry, and mobile robotics, with the aim of improving robustness and efficiency [25,27,28]. Registration methods commonly follow a coarse-to-fine strategy and employ different types of geometric features for registration. Geometric features, such as points, lines, and planes, are often used in TLS point cloud coarse registration due to their strong geometric constraints in urban environments. Point-feature-based registration methods rely on algorithms such as 3D SURF [29], Harris 3D [30], RANSAC [31], and the 4PCS-based method [32] for feature detection. However, these methods can be sensitive to variations in point cloud density and noise [33]. Line features are often used in urban TLS point cloud registration since roads and buildings, which are the most typical infrastructure in urban space, provide unique line features [34]. Al-Durgham and Habib [35] insist that line features have better geometric constraints than point features and yield more accurate results. Prokop et al. [36] proposed a line-feature-based solution for a very low overlapping indoor area between the two scans. However, these approaches still exhibit low accuracy and slow computation speed relative to other methods.

Plane-feature-based coarse registration methods are less affected by point cloud density and noise, making them suitable for various types of input point clouds [37,38,39]. Theiler and Schindler [40] proposed a coarse registration method using indirect plane features, utilizing virtual tie points generated by intersecting planar surfaces in the scene. Xu et al. [41] introduced a semi-automatic coarse registration approach for point clouds using the geometric constraints of voxel-based 4-plane congruent sets. Inspired by these methods, several automatic coarse registration methods based on plane features have been developed [42,43]. Chen et al. [44] utilized plane and line extraction for registering scans with limited features and small overlap in arbitrary initial poses. Zou et al. [45] utilized integrated navigation system (INS) sensors to acquire longitude, latitude, altitude, and attitude data for their mobile mapping systems (MMS). They then conducted a plane-based global registration using this data to perform 3D reconstruction. In summary, plane-feature-based approaches offer robustness and good performance in urban environments with man-made artifacts [26].

With the development of AI technology, several coarse registration methods based on deep learning have also been proposed. Zeng et al. [46] introduced a voxelization-based registration method called 3DMatch that divides point clouds into regular 3D grids and utilizes a 3D convolution network structure. Ao et al. [47] presented a deep-learning-based surface descriptor called SpinNet for 3D point cloud registration. Li et al. [48] designed the point cloud multi-view convolution neural network (PC-MVCNN) for point cloud registration. However, deep-learning-based approaches face a critical limitation due to their dependence on the training dataset. Facilities exhibit diverse structures and shapes depending on the purpose of the building or infrastructure. Therefore, a model trained with a general facility might fail to perform segmentation of the point cloud successfully for different types of sites [10,49]. Additionally, the limited amount of input data and complexity pose challenges in applying these methods to large-scale point clouds [26].

The Iterative Closest Point (ICP) algorithm [50] is the most commonly used method for fine registration [25,26]. Variants of the ICP algorithm, including point-to-plane ICP [51] and plane-to-plane ICP [52], have also been developed. Zhang et al. [53] proposed a new ICP method to improve the convergence speed and robustness of point-to-point and point-to-plane ICP methods. However, despite their popularity, those ICPs and their variants heavily rely on a good initial estimation and the quality of the point cloud data, considering factors like noise, density, and occlusions.

Another frequently employed technique for fine registration is the normal distribution transform (NDT) algorithm [54]. NDT-based registration methods represent point cloud data using probability density functions (PDF) and assess the similarity between these functions. Several variants of NDT have been proposed, involving differentiating the grid size for PDFs [55,56] and evaluating similarity through point-to-distribution or distribution-to-distribution approaches [57]. However, NDT and its variants face similar limitations to ICP, as they also depend on a reliable initial estimation.

#### 1.2.2. Point Cloud Registration for Stop-and-Go Scanning System

To address the static nature of traditional TLS, many various moving platforms have been explored to develop stop-and-go scanning systems. Blaer and Allen [20] pioneered the stop-and-go scanning system using an unmanned ground vehicle (UGV) and optimized its scanning plan. Notably, they relied on manual registration using reflective targets distributed throughout the environment. Kurazume et al. [58] devised a TLS scanning system that employed multiple UGVs, which included a primary robot and subsidiary robots. Intriguingly, the child robots were utilized as target markers for the registration process. Liu et al. [59] introduced a vehicle-based stop-and-go scanning system tailored for detecting deformations in highway bridgeheads. For registration, they manually selected the shared feature points between two scans, such as traffic sign corner points and the guardrail cylinder center. Additionally, some researchers have investigated innovative robot models. For instance, Park et al. [16] launched an automated stop-and-go scanning system powered by a quadruped walking robot, Boston Dynamics Spot. However, the post-scan registration was still manually executed. The recurring theme of manual and target-based registration processes underlines their inefficiency and the need for an automated solution. Most of these systems either overlook the imperative of automatic registration or remain anchored to laborious manual registration techniques [16,20,58,59]. This oversight not only introduces inefficiencies but also stretches the time required for post-acquisition processing.

To address this issue, several researchers have suggested automatic registration techniques. These methods are designed to potentially integrate with stop-and-go scanning systems, and many have leaned on supplementary data-guided registration using additional sensor data, such as odometry, inertial measurement units (IMU), and global Navigation satellite system (GNSS), a camera, and a robot, to improve the registration process [60,61,62]. Lin et al. [63] conceptualized a MMS framework grounded in the TLS-based stop-and-go mapping concept. The proposed registration relied on scanner poses derived from IMU/GPS data and the ICP algorithm. Yet, only a theoretical analysis was offered without testing in real-world scenarios. A distinct approach taps into visual sensor image data. Mohammed et al. [64] deployed RGB images linked with scans, utilizing the SURF algorithm to estimate relative orientation between scans. Ge et al. [61] described an image-informed, end-to-end registration strategy for unordered TLS point cloud datasets. Recent methodologies have incorporated scan planning data. Knechtel et al. [65] presented a 2D polygon-centric optimal scan planning technique for the Boston Dynamics Spot and crafted a connectivity graph derived from the planning results. This graph informs the ICP-based registration process. Although these studies did not validate their methods with tangible stop-and-go scanning systems, they showcased their significant potential for remedying the auto-registration challenges inherent in stop-and-go scanning applications. Foryś et al. [66] introduced the FAMFR algorithm to precisely register two-point clouds representing various cultural heritage interiors based on two different handcrafted features, utilizing the color and shape of the object to accurately register point clouds with extensive surface geometry details or geometrically deficient but with rich color decorations.

Recent endeavors have moved towards implementing automated registration approaches for real-world stop-and-go scanning systems. Chow et al. [21] revealed a continuous stop-and-go scanning system that combined IMU and RGB-D cameras for registration. However, the system was based on a rudimentary wheeled handcart requiring manual operation. Borrmann et al. [67] constructed thermal 3D geometric data rooted in a mobile robot’s odometry readings. The simultaneous localization and mapping (SLAM) algorithm determined the scanner’s position. Zhong et al. [60] rolled out a stop-and-go scanning system for rural cadastral assessments using a vehicle-backed MMS and GNSS-IMU sensors. Prieto et al. [23] addressed the registration conundrum by harnessing the mobile robot’s odometry and the trusted ICP method. The visibility analysis conducted for their next best scan (NBS) algorithm is based on a 2D vertical image projection. This approach involves vertically projecting the point cloud onto each 3D model of a wall component and calculating the visibility from the number of pixel points and the overall size of the image. However, these studies’ primary focus was not on refining the registration technique. Consequently, geometric data accuracy and computational performance remained unevaluated.

### 1.3. Existing Problems and Research Objectives

Table 1 summarizes the literature review on the stop-and-go scanning system in terms of its registration method. The recent trend in automatic point cloud registration with the stop-and-go scanning system involves the use of prior information from additional sensors or the mobile platform itself. However, the major limitations of previous research include: (1) Many studies have not explored automatic registration for their systems, even though some researchers have proposed theoretical automatic registration methods for the stop-and-go scanning system. Some have addressed this, but recent studies conducted with state-of-the-art mobile platforms are limited; (2) The majority of the registration processes assume a high overlapping ratio between two point clouds. This presumption restricts the applicability of these methods in irregular scenarios with low overlap scan data; (3) While a few studies have tackled automatic registration with autonomous data acquisition using robots, they often overlook the evaluation of their registration processes. Consequently, the quality of their complete point cloud data remains unverified.

To address these challenges, this study introduces an automated point cloud registration approach optimized for a stop-and-go scanning system. This approach efficiently registers 3D scan data using mobile platform localization as prior knowledge. The objectives of the study are as follows:To present an automatic point cloud registration approach for the stop-and-go scanning system considering plane-based registration, which uses prior information from the localization of the mobile platform;To introduce a robust method capable of handling irregular scenarios with low overlap scan data;To evaluate the proposed method and compare it with conventional registration methods;Substantiation of the proposed method with the stop-and-go scanning system using the state-of-the-art mobile platform, quadruped walking robot in real-world indoor environments.

The remainder of this chapter is structured as follows: Section 2 proposes the automatic point cloud registration method. Section 3 presents the implementation of this method, accompanied by the results of point cloud registration from real-world experimental scans. Section 4 discusses the strengths and limitations of the approach. Finally, in Section 5, the conclusion and future works are presented.

## 2. Methodology

### 2.1. Overview

The general concept of registration for TLS scan data is to align multiple point clouds from local coordinate systems into a unified coordinate system. This involves determining 3D rotation and translation values for each local point cloud, taking into account their geometrical relation. In this study, the focus is on achieving registration by utilizing information from the mobile platform, which is well-suited for scenarios with low overlapping scans in indoor environments. To facilitate the registration process, two key assumptions are made:The approximate locations of TLS are known: It is assumed that the locations of the TLS scans are approximately known, and therefore, consideration for translations is not required during the coarse registration phase. This assumption is valid for most state-of-the-art mobile platforms, as they come equipped with their own localization function, enabling accurate location information for scan positions.Rotation around the *z*-axis is sufficient for registration. During the registration, only the rotation around the *z*-axis is needed to determine the correct rotations of the scans relative to each other. This assumption is reasonable because modern LiDAR scanners are typically equipped with a gyroscope sensor, and their output is correctly aligned with the gravity vector, pointing in the negative direction of the *z*-axis [36].

The proposed method comprises three main phases: (1) the initial phase begins with the extraction of perpendicular constrained wall planes from the input point cloud, sourced from the stop-and-go scanning system detailed in our previous study [16]. The point cloud is first subjected to voxel filtering and then downsampled. Subsequently, points presumed to be parts of walls are identified based on their normal vectors. Using the RANSAC-based algorithm, M-SAC wall-plane models adhering to perpendicular constraints are extracted; (2) the second phase involves the alignment of the wall-planes, taking into account the scan system’s initial position. Potential rotation angles are assessed by comparing the normal vector alignment of each wall plane. Overlapping scores for each angle are then computed using a point-to-point displacement analysis. The modifiable-nest-octree-(MNO)-based point cloud indexing technique is utilized to enhance computational efficiency, aiding in the accurate calculation of point-to-point displacements. This computation marks the completion of the coarse registration process and determines the matching patches essential for the next fine registration phase; (3) the final stage focuses on the precise registration of the identified matching patches. A horizontality constrained ICP algorithm is proposed and used for this purpose. The proposed method is automated in all of the phases of the workflow without any manual intervention. Figure 1 illustrates the procedure for the proposed method.

### 2.2. Perpendicular Constrained Wall-Plane Extraction

Many traditional point cloud registration methods focus on identifying salient features and use local geometric descriptors to establish correspondences between point clouds. However, ensuring sufficient overlap and descriptive features can be challenging in an indoor environment [44]. Therefore, this approach aims to extract wall planes as geometric features, taking advantage of the fact that man-made environments often consist of planar structures, such as walls. These walls represent the main structure of the scene and can provide valuable information for robust registration [68]. Additionally, a constrained condition that adjusts wall-plane extractions was applied based on the assumption that all such walls are parallel or perpendicular to each other [6,7].

#### 2.2.1. Voxel-Based Filtering and Alignment along the Z-Axis Subsubsection

The point cloud data acquired by TLS at an extensive indoor site exhibits the following characteristics: Firstly, their density is irregular, with points closer to the sensor being densely distributed, while those farther away are sparsely represented. Secondly, the data volume is substantial, with a single station’s TLS data comprising several million points in sizable scenarios, demanding significant computational resources [48]. To address this issue, several studies have adopted voxel-based filtering for point cloud downsampling [28,42,48,69] because the subsampled point cloud has a more even point density and lower point count, which benefits the efficiency of registration [28,69]. Voxel filtering is used to preprocess the point cloud, reducing the influence of noise points on registration accuracy and improving efficiency. It is insensitive to density and can remove noise points and outliers while downsampling the point cloud to some extent. Therefore, it is adopted in this study as input for coarse registration. The spatial resolution for voxel-based filtering: based on the maximum construction tolerance standard from the Korean Construction Standard Specification (KCS) [70], the value was selected as 0.05 m.

Following the voxel filtering process, the fine alignment of the normal vector of the ceiling and floor along the *Z*-axis is conducted. Although modern LiDAR scanners are correctly aligned with the gravity vector, pointing in the negative direction of the *Z*-axis, there may be some fine misalignment due to the vibration of the robot system. The main XY plane of the input point cloud is extracted using the M-SAC algorithm [71], which is an extension of the RANSAC algorithm [72], within the normal vector under a given threshold angular of 5 degrees to the *Z*-axis. The rotation parameters to align the normal vector with the *Z*-axis are estimated, and the input point cloud is transformed using these rotation parameters.

#### 2.2.2. Estimation of Wall Point Cloud with Normal Vector Calculation

For the robust fitting of the wall-plane model, this section outlines the initial estimation procedure applied to identify points that are estimated to be part of the wall component. Given that most walls within indoor scenes exhibit verticality relative to the ground floor, the points constituting the wall structure should have normal vectors that are perpendicular to the normal vector of the ground plane. Essentially, this implies that the z component of the unit normal vector should be close to 0.

The estimation of normal for individual points can be achieved by considering neighboring points to determine a local plane [73]. In the interest of both safety and computational efficiency, this study opts to utilize four neighboring points to compute each normal vector. Let 1, 2, …, and s be the labels of the points and $p_{s}\in P$ be a point where P denotes the set of all points from the scan point cloud. Also, let $u_{s}\in U$ be a unit vector of $p_{s}$, where U denotes a set of all unit vectors from P. Thus, $w_{P}$, the set of segmented wall points from P, is mathematically expressed in Equation (1).
(1)$$w_{P}=\left\{p_{s}\in P\;|\;\sqrt{{\left(u_{s_{x}}\right)}^{2}+{\left(u_{s_{y}}\right)}^{2}}≅1\right\}$$The terms are defined as follows:
$w_{p_{s}}$: a set of segmented points as wall,$u_{s_{x}},u_{s_{y}}:x,y\mathrm{components}\;\mathrm{of}\;u_{s}\;\mathrm{in}\;a\;\mathrm{Cartesian}\;\mathrm{coordinate}\;\mathrm{system}$

Figure 2 illustrates the process of estimating the wall point cloud. In Figure 2a, the input point cloud is visualized, comprising a variety of points. In Figure 2b, the estimated wall point cloud is presented, with wall structures highlighted. However, there are also some noise points present that do not belong to the walls.

#### 2.2.3. RANSAC-Based Wall Extraction with Perpendicular Constraint

After the estimation of the wall point cloud in the previous step, it was observed that some noise points were also included, as depicted in Figure 2b. To robustly model the wall plane, the M-SAC algorithm [71], which is an extension of the RANSAC algorithm [72], is adopted for the segmented point cloud. In the RANSAC approach, fitting is considered successful if inlier points exist within the specified threshold, without further assessing how close these points are to the desired plane within the threshold. However, the M-SAC algorithm evaluates the sum of the distances between all points and the model. In essence, it utilizes a quality measure that considers how closely the inlier points align with the desired model. This refined criterion allows for a more accurate and reliable assessment of the model fitting process. In this study, the threshold distance was established at 0.05 m, considering the construction tolerance standard stipulated by the Korean Construction Standard Specification [70].

The majority of indoor building environments adhere to the Manhattan world condition, characterized by geometric properties such as orthogonality or parallelism among wall structures. Consequently, during the wall-plane extraction process, these perpendicular constraints were imposed to identify robust matching features. The initial wall plane was first extracted from the segmented point cloud using the M-SAC algorithm. The normal vector of this initial wall plane was designated as the reference vector. Subsequently, a direction vector perpendicular to the reference vector along the *z*-axis was computed, and the perpendicular wall plane was derived using this direction vector. This iterative process continued until the number of points within the extracted plane fell below 1000. For clear separation of the walls, a minimum distance of 1 m was maintained between each extracted plane. The wall-plane extraction process using the M-SAC algorithm can be summarized using Algorithm 1.

**Algorithm 1:** The wall-plane extraction with the M-SAC algorithm# **Definition 1**: ‘$w_{P}$’ is defined as the set of segmented wall points.# **Definition 2**: ‘*plane_refer*’ is defined as a set of wall planes with the reference vector.# **Definition 3**: ‘*plane_ortho*’ is defined as a set of the perpendicular wall planes.**Input:** $w_{P}$**Output:** *plane_refer, plane_ortho*1: *plane_refer* = empty array;2: *plane_ortho* = empty array;3: $[\mathrm{initial}\text{\_}\mathrm{plane},\;\mathrm{remain}\text{\_}w_{P}$$]=\text{M-SAC}(w_{P})$;4: **add** *initial_plane* **to** *plane_refer*5: *reference_vector* = XY plane-orthogonal projection vector of *initial_plane.normal;*6: *reference_ortho_vector* = orthogonal vector of *reference_vector;*7: $[\mathrm{initial}\text{\_}\mathrm{ortho}\text{\_}\mathrm{plane},\;\mathrm{remain}\text{\_}w_{P}$$]=\text{M-SAC}(remain\_w_{P}$, *reference_ortho_vector*);8: **add** *initial_ortho_plane* **to** *plane_ortho*9: **while** (true);10: $\;[\mathrm{temp}\text{\_}\mathrm{plane},\;\text{\_}w_{P}$$]=\text{M-SAC}(remain\_w_{P}$, *reference_vector*);11: $\;\mathrm{if}\;\mathrm{length}(\mathrm{remain}\_w_{P}$) < 100012:                 **break**13:     **end**14:     **if** all distances between *temp_plane* and planes of *plane_refer* > 115:                 **add** *temp_plane* **to** *plane_refer*16:     **end**17: **end**18: **while** (true);19:    $\;[\mathrm{temp}\text{\_}\mathrm{ortho}\text{\_}\mathrm{plane},\;\mathrm{remain}\text{\_}w_{P}$$]=M-\mathrm{SAC}(remain\_w_{P}$, *reference_ortho_vector*);20:     $\mathrm{if}\;\mathrm{length}(\mathrm{remain}\text{\_}w_{P}$) < 100021:                 **break**22:     **end**23:     **if** all distances between *temp_ortho_plane* and planes of *plane_ortho* > 124:                 **add** *temp_ortho_plane* **to** *plane_ortho*25:     **end**26: **end**27: **return** *plane_refer, plane_ortho*

Figure 3 depicts the M-SAC-based wall extraction process. In Figure 3a, the input point cloud is visualized, having been segmented in the preceding section. Figure 3b presents the extracted wall-point cloud, with non-wall points distinctly removed.

### 2.3. Coarse Registration with Plane Matching

After extracting the wall planes from both the reference and target point clouds, the subsequent task involves matching the target planes with the reference planes. Potential rotation angles for coarse registration are determined by assessing the alignment of normal vectors of each wall plane. For each potential rotation angle, overlapping scores are computed through point-to-point displacement. The coarse registration is then confirmed using the rotation angle that yields the highest score, and matching patches are identified for the subsequent fine registration. In this process, an MNO-based point cloud indexing approach is employed to minimize computational demands.

#### 2.3.1. Calculation of Rotation Angles for the Coarse Registration

Based on the assumption that all walls are either parallel or perpendicular to each other, a set of rotation candidates, four rotation angles around the *Z* axis, is proposed for the target wall plane relative to the reference wall plane. In Figure 4a, a top view displays the reference wall planes in red and the target wall planes in blue. Since all target wall planes are either parallel or perpendicular to one another, calculating angular transformations is simplified. This is achieved by aligning a normal vector from one target wall plane to the normal vectors of the two reference wall planes. From the target wall plane, two opposing cases of the normal vector can be estimated. This leads to four distinct angular relationship scenarios, as depicted in Figure 4b. As a result, four temporary transformed target point clouds (green) are generated based on the set of rotation candidates (Figure 4c).

#### 2.3.2. MNO-Based Point Cloud Indexing

To determine the rotation angle for coarse registration from the angle candidates, a point-to-point displacement-based overlapping score was employed, as detailed in Section 2.3.3. However, calculating the displacement between two-point clouds demands substantial computational resources, as it involves processing each point in both point clouds individually. To tackle this challenge, a data structure approach based on the modifiable nested octree (MNO) indexing method was proposed.

Several research works have illustrated the impact of the point cloud’s data structure on computational efficiency [10,15]. MNO, a data structure for point clouds, has been recognized as an effective indexing method for handling and visualizing extensive point clouds [74,75]. Leveraging these attributes, this study utilized an MNO structure based on the point cloud during the point-to-point displacement computation. The MNO indexing is initially conducted with $w_{P}$ of the reference point cloud (Figure 2b). Figure 5 exemplifies MNO indexing [10].

#### 2.3.3. Overlapping Score with Point-to-Point Displacement

During this section, a data structure-based point-to-point displacement calculation was employed to compute overlapping scores. This computation concludes the coarse registration process and identifies the matching patches necessary for the subsequent fine registration. The target point cloud is indexed within the pre-existing MNO of the reference point cloud, established in the previous section. This indexing facilitates the computation of displacements for points within each individual MNO node in both the target and reference clouds, resulting in a significant reduction in computation time [10,76]. Figure 6 shows the indexing process of the target point cloud with the constructed MNO structure. The target point cloud (Figure 6a) is located on the MNO structure (Figure 6b), and the points located in each MNO node index to that node. Figure 6c presents the indexed target point cloud. The points within the same MNO node are depicted in the same color, while the points out of the MNO structure are depicted in black.

Following the indexing process, the target and reference points within each individual MNO node are retrieved, and their displacements are calculated. In this research, displacement is defined as the distance between a point in the first point cloud and the closest point in the second point cloud to that point. Once the displacements of all points within each MNO node are computed, points with displacements surpassing the user-defined threshold are eliminated (lines 15 and 18 in Algorithm 2). Subsequently, the overlapping score is derived by comparing the ratio of remaining points (considered as overlapping points) to the initial number of points. Lines 21–23 in Algorithm 2 briefly describe this process.

**Algorithm 2:** Calculation of the overlapping score in the MNO structure# **Definition 4**:$\;‘\mathrm{reference}\_w_{P}$$’\;\mathrm{is}\;\mathrm{defined}\;\mathrm{as}\;\mathrm{indexed}\;w_{P}$ of the reference point cloud.# **Definition 5**:$\;‘\mathrm{target}\_w_{P}$$’\;\mathrm{is}\;\mathrm{defined}\;\mathrm{as}\;\mathrm{indexed}\;w_{P}$ of the target point cloud.# **Definition 6**: ‘*threshold*’ is defined as a user-defined threshold.# **Definition 7**: ‘*overlap_score*’ is defined as the overlapping score.# **Definition 8**: ‘displacement2point’ is a function for calculating the displacement.**Input:** *reference_*$w_{P}$*, target_*$w_{P}$*, threshold***Output:** *overlap_score*1:     *pc_refer_overlap* = empty array;2:     *pc_target_overlap* = empty array;3:     *overlap_score* = empty array;4:     *binNums* = index of MNO nodes;5:     **for** i = *binNums* **do**6:                 $\mathrm{temp}\_\mathrm{refer}\_\mathrm{point}=\mathrm{points}\;\mathrm{of}\;\mathrm{reference}\_w_{P}$ in MNO node(i)7:                 **if** *temp_refer_point* is empty **then**8:                           **continue**9:                 **end**10:               $\mathrm{temp}\_\mathrm{target}\_\mathrm{point}=\mathrm{points}\;\mathrm{of}\;\mathrm{target}\_w_{P}$ in MNO node(i)11:     **if** *temp_refer_point* is empty **then**12:                 **continue**13:     **end**14:     *temp_refer_dis* = displacement2point(*temp_refer_point*, *temp_target_point*);15:     *temp_refer_in* = points of displacements within *threshold* in *temp_refer_point;*16:     **add** *temp_refer_in* **to** *pc_refer_overlap*17:     *temp_target_dis* = displacement2point(*temp_target_point, temp_refer_point*);18:     *temp_target_in* = points of displacements within *threshold* in *temp_target_point;*19:     **add** *temp_target_in* **to** *pc_target_overlap*20:     **end**21:     *num_in* = length(*pc_refer_overlap) +* length(*pc_target_overlap);*22:     *num_total* = length(*temp_refer_point) +* length(*temp_target_point);*23:     *overlap_score* = *num_in/num_total;*24: **return** *overlap_score*

After calculating the overlapping scores for each potential rotation angle, the rotation angle with the highest score is chosen as the final transformation for the coarse registration. Figure 7 illustrates an example of the final step of coarse registration with the overlapping score. In Figure 7a, you can see the reference point cloud (red) and the target point cloud (blue) as examples. Figure 7b,d,f,h shows the results (green) after applying each potential rotation angle. Correspondingly, Figure 7c,e,g,i shows the overlapping points within a displacement threshold of 0.5 m, between the reference point cloud (red) and the transformed target point cloud (green) pairs from Figure 7b,d,f,h. It is clear that the pair shown in Figure 7b,c has the most overlapping points compared to the other pairs, resulting in the highest overlapping score. As a result, the scenario shown in Figure 7b is selected as the coarse registration outcome, and the overlapping points (Figure 7c) are identified as the matching patches for the subsequent fine registration. Figure 7j provides a visual representation of the coarse registration outcome in this example.

### 2.4. Fine Registration with Horizontality Constrained ICP

For the fine registration, the point-to-plane ICP algorithm [77,78] is employed to achieve a precise estimation of the registration parameters using the coarse registered point cloud. The fundamental concept involves minimizing point-to-plane correspondences, which relate to the sum of the squared distances between a point and the tangent plane at its corresponding point, and determining the rigid transformation parameters between these points [64,79]. This process can be viewed as an optimization challenge that aims to minimize the error by reducing the cumulative Euclidean distances between the sets of points and corresponding planes. The objective function for each ICP iteration is to find $M_{opt}$, the optimized rigid transformation matrix, which is mathematically expressed in Equation (2) [77,78].
(2)$$M_{opt}=ARG\;{min}_{M}\sum_{i}{\left((M\cdot {tp}_{i}-{rp}_{i})•n_{i}\right)}^{2}$$
The terms are defined as follows:
${M\;and\;M}_{opt}:\;\mathrm{the}\;4\;\times \;4\;3D\;\text{rigid-body}\;\mathrm{transformation}\;\mathrm{matrices},$${tp}_{i}\;:\;{\left({tp}_{ix},{tp}_{iy},{tp}_{iz},1\right)}^{t},\;a\;\mathrm{target}\;\mathrm{point}\;\mathrm{cloud},$${rp}_{i}:{\left({rp}_{ix},{rp}_{iy},{rp}_{iz},1\right)}^{t},\;a\;\mathrm{reference}\;\mathrm{point}\;\mathrm{cloud},$$n_{i}:\;{\left(n_{ix},\;n_{iy},\;n_{iz},\;1\right)}^{t},\;\mathrm{the}\;\mathrm{unit}\;\mathrm{normal}\;\mathrm{vector}\;\mathrm{at}\;{rp}_{i}.$

However, this study operates under the assumption that only the rotation around the *z*-axis is considered. Under this assumption, a 3D rigid-body transformation M is composed of a rotation matrix R(0,0,γ) and a translation matrix $T(t_{x},\;t_{y},\;t_{z})$,
(3)$$M=T(t_{x},\;t_{y},\;t_{z})\cdot R(0,0,\gamma )$$
where
(4)$$T(t_{x},\;t_{y},\;t_{z})=\left(\begin{array}{cccc} 1 & 0 & 0 & t_{x}\\ 0 & 1 & 0 & t_{y}\\ 0 & 0 & 1 & t_{z}\\ 0 & 0 & 0 & 1 \end{array}\right)$$

And
(5)$$\begin{array}{l} R\left(0,0,\gamma \right)=R_{z}\left(\gamma \right)\cdot R_{y}\left(0\right){\cdot R}_{x}\left(0\right)\\ =\left(\begin{array}{cccc} \cos\gamma & -\sin\gamma & 0 & 0\\ \sin\gamma & \cos\gamma & 0 & 0\\ 0 & 0 & 1 & 0\\ 0 & 0 & 0 & 1 \end{array}\right) \end{array}$$
$R_{x}\left(0\right)$, $R_{y}\left(0\right)$ and $R_{z}\left(\gamma \right)$ are rotations of 0, 0 and γ radians about the x-axis, y-axis and z-axis, respectively.

Equation (2) represents a least-squares optimization problem, the solution of which necessitates determining the values of four parameters: γ, $t_{x}$, $t_{y}$, and $t_{z}$.
(6)$$\begin{array}{l} M=T\left(t_{x},\;t_{y},\;t_{z}\right)\cdot R\left(0,0,\gamma \right)\\ =\left(\begin{array}{cccc} \cos\gamma & -\sin\gamma & 0 & t_{x}\\ \sin\gamma & \cos\gamma & 0 & t_{y}\\ 0 & 0 & 1 & t_{z}\\ 0 & 0 & 0 & 1 \end{array}\right) \end{array}$$

Subsequently, Equation (2) can be reformulated as a nonlinear equation involving the four parameters γ, $t_{x}$, $t_{y}$, and $t_{z}$.
(7)$$F_{i}(\gamma ,\;t_{x},t_{y},t_{z})=\left(M\cdot {tp}_{i}-{rp}_{i}\right)•n_{i}=\left(M\cdot \left(\begin{array}{c} {tp}_{ix}\\ \;{tp}_{iy}\\ {tp}_{iz}\\ 1 \end{array}\right)-\left(\begin{array}{c} {rp}_{ix}\\ \;{rp}_{iy}\\ \;{rp}_{iz}\\ 1 \end{array}\right)\right)•\left(\begin{array}{c} n_{ix}\\ \;n_{iy}\\ \;n_{iz}\\ 0 \end{array}\right)\;=\left[\left(n_{ix}{tp}_{ix}+n_{iy}{tp}_{iy}\right)\cos\gamma +\left({-n_{ix}{tp}_{iy}+n}_{iy}{tp}_{iy}\right)\sin\gamma +n_{ix}t_{x}+n_{iy}t_{y}+n_{iz}t_{z}\right]\;-\left[n_{ix}{rp}_{ix}+n_{iy}{rp}_{iy}+\;n_{iz}{rp}_{iz}-n_{iz}{tp}_{iz}\;\right]$$

Given *N* pair of point correspondences, all nonlinear equations can be linearized using a Taylor series approximation, which can be expressed as follows [80]:(8)J·X=K+V
where
(9)$$K=\left(\begin{array}{c} F_{1}\left(\gamma ,\;t_{x},t_{y},t_{z}\right)-F_{1}\left(\gamma_{0},\;t_{x_{0}},t_{y_{0}},t_{z_{0}}\right)\\ F_{2}\left(\gamma ,\;t_{x},t_{y},t_{z}\right)-F_{2}\left(\gamma_{0},\;t_{x_{0}},t_{y_{0}},t_{z_{0}}\right)\\ \vdots \\ F_{N}\left(\gamma ,\;t_{x},t_{y},t_{z}\right)-F_{N}\left(\gamma_{0},\;t_{x_{0}},t_{y_{0}},t_{z_{0}}\right) \end{array}\right)$$
(10)$$X=\left(\begin{array}{c} d\gamma \\ dt_{x}\\ dt_{y}\\ dt_{z} \end{array}\right)\;\;\;\;\;\;\;\;\;V=\left(\begin{array}{c} v_{1}\\ v_{2}\\ \vdots \\ v_{N} \end{array}\right)$$

With
(11)$$J=\left(\begin{array}{cccc} \frac{\partial F_{1}}{\partial \gamma } & \frac{\partial F_{1}}{\partial t_{x}} & \frac{\partial F_{1}}{\partial t_{y}} & \frac{\partial F_{1}}{\partial t_{z}}\\ \frac{\partial F_{2}}{\partial \gamma } & \frac{\partial F_{2}}{\partial t_{x}} & \frac{\partial F_{2}}{\partial t_{y}} & \frac{\partial F_{2}}{\partial t_{z}}\\ \vdots & \vdots & \vdots & \vdots \\ \frac{\partial F_{N}}{\partial \gamma } & \frac{\partial F_{N}}{\partial t_{x}} & \frac{\partial F_{N}}{\partial t_{y}} & \frac{\partial F_{N}}{\partial t_{z}} \end{array}\right)$$

In this study, the Levenberg-Marquardt optimization method is employed. Consequently, the vector of least-squares corrections in the system of equations is given by:(12)$$X={\left(J^{T}\cdot J+\lambda \cdot diag(J^{T}\cdot J)\right)}^{-1}\cdot J^{T}\cdot K$$
The term is defined as a follow:
λ: a damping factor

However, if the overlap between two point clouds is insufficient, the ICP algorithm’s objective of minimizing error by reducing cumulative Euclidean distances among sets of points and correspondeding planes will not be effectively achieved. This is because the algorithm optimizes the error for all points, and inadequate overlap can lead to inaccurate registration results. To address this, matching patches from the previous phase can be used. In Figure 8a,b, an example of horizontality constrained ICP registration result using a point cloud with low overlap is shown, where the registration has generally failed. However, with the application of the algorithm to the matching patches, as depicted in Figure 8c,d, precise registration results are obtained. The final registration parameters are then applied to the target point cloud, as shown in Figure 8e, demonstrating the successful outcome of the fine registration. The registration process was iterated for all scan point clouds in accordance with the order of scanning performed by the stop-and-go scanning system.

## 3. Experimental Results

To verify the feasibility of the proposed approach, experiments were conducted in three steps: (1) data acquisition using a stop-and-go scanning system [16] in a real indoor environment; (2) registration process with the proposed approach and validation of its performance; (3) performance comparison with the conventional approach and several benchmark datasets. Experiments for the registration process were performed using MATLAB 2023b on a system with an AMD Ryzen 7 3700X CPU @3.59 GHz processor and 16 GB of RAM.

### 3.1. Test Site

Baekyangnuri, situated at Yonsei University in the Republic of Korea, encompasses diverse research and educational support facilities, parking zones, and cultural spaces. For the test site, a common corridor was chosen due to its extensive indoor expanse, necessitating multiple scan points. This corridor typifies a structure with elongated passages, incorporating elements like windows, doors, and stairs. Notably, the corridor layout also includes circular segments, with floor heights varying across different sections (depicted in Figure 9).

### 3.2. Data Collection with a Stop-and-Go Scanning System

In this study, data acquisition was carried out using the stop-and-go scanning system and developed in the previous study [16]. The quadruped walking robot, Boston Dynamics Spot [81], was employed to the stop-and-go scanning system. It was equipped with a MLS, Velodyne VLP [82], to enhance the robot localization, and a TLS, FARO Focus 3D laser scanner [83], which is capable of rectifying auto-leveling discrepancies up to 2 degrees of misalignment for precise 3D mapping, as depicted in Figure 10. The FARO scanner was wirelessly connected to the robot system via a fixed IP. Boston Dynamics Spot is a quadrupedal robot designed for mobility across challenging terrains and rugged surfaces, including steps and stairs [16,84]. Notably, it features a robust obstacle avoidance system that encompasses both fixed and dynamic obstacles. This system provides the robot with a broad navigational space, allowing for flexible safety constraints such as safe distances and floor slopes. The parameter values for the stop-and-go scanning system in this study are summarized in Table 2. Regarding the numbering and location of scan positions, these were determined based on our previous research on scan planning [16]. The optimal scan positions were calculated through a 3D visibility analysis involving ray tracing of the BIM geometry, and the number of scan positions was selected using the proposed optimization algorithm.

### 3.3. Registration Results

Before initiating the registration process, it is imperative to unify the coordinate systems of both the scanning system and the reference point cloud, with the latter serving as the starting point of the registration process. Subsequently, a fundamental surveying process was conducted using a total station to acquire control points for the real-world test site. Figure 11a illustrates the distribution of control points (CP) across the test site in relation to the reference point cloud, which is numbered as 1. For 3D coordinate transformation, a minimum of three CPs are mandatory. In this study, seven CPs were surveyed and employed for least squares adjustment to enhance the transformation performance. The locations of these CPs were strategically chosen to be easily identifiable by the total station, such as the corners of columns, doors, and walls. A total of 7 CPs were surveyed, and the coordinate systems of the scanning system and the reference point cloud are unified using these CPs. Subsequently, prior location information from the scanning system and the reference point cloud became available for the proposed registration approach.

Figure 12 depicts the registration result of the proposed approach. The colors within the point clouds represent the variations among each scan station. This depiction validates the effectiveness of the proposed method in handling expansive scenes without encountering registration failures.

To conduct a quantitative accuracy analysis, a fundamental surveying process was executed using a total station to validate the quality and reliability of the registered point cloud data. The accuracy assessment was conducted following the method proposed in [6,7], wherein well-distributed and easily identifiable points acquired from a highly accurate total station serve as reference targets for comparison with corresponding points obtained from the registered point cloud data. Figure 13a illustrates the distribution of target points (TP) across the test site. Red dotted circles indicate the locations of assessment targets. Figure 13b,c provides both a real-world view and point cloud data of the shaded circle. Identifiable target points, such as column corners, doors, and walls, were acquired from both the total station and the registered point cloud data. It should be noted that a minimum of three TPs were obtained for each scan station. The positional accuracy was evaluated using the root mean square error (RMSE) as defined in Equation (13).
(13)$$RMSE=\sqrt{\frac{1}{n}\sum_{i}^{n}{\left(\left(P_{i}^{g}-(R\cdot P_{i}^{l}+t\right)\right)}^{2}}$$The terms are defined as follows:
$P_{i}^{g}:\;a\;i^{th}\;target\;point\;in\;global\;coordinate\;system\;from\;total\;station,$$P_{i}^{l}\;:\;a\;i^{th}\;target\;point\;in\;local\;coordinate\;system\;from\;registered\;point\;cloud,$$R\in R^{3\times 3}:\;the\;rotation\;matrix\;of\;the\;registration\;parameters,$$t\in R^{3}:\;the\;translation\;matrix\;of\;the\;registration\;parameters$

A total of 35 target points were used for validation. For example, the numbers in red circles in Figure 13b,c indicate the target number. The calculated RMSE value was 0.044 m, underscoring the performance of the proposed method, especially considering that the spatial resolution of sampling in this study is 0.05 m.

Table 3 presents details regarding the collected scan data, computation time, and the successful registration rate (SRR). The SSR is an evaluation metric utilized in various registration studies [28,41]. *σ**_p_* was used to represent the predefined threshold (in this study *σ**_p_* = 0.1 m) for position error of surveying points, $e^{p}$, in each scan point cloud. Successful registration (SR) for each scan is then calculated by Equation (14).
(14)$$SR=\{\begin{array}{cc} \;1 & e^{p}\;<\;\sigma_{p}\\ 0 & \;\mathrm{otherwise}\; \end{array}$$

So, the SRR is defined by Equation (15).
(15)$$SRR=\frac{N_{s}}{N-1}$$
where $N_{s}$ is the number is the number of SR and *N* is the number of total scans.

The total point count amounted to 186 million. The registration process took 424.2 s to complete. For the overlapping ratio, the overlapping score values outlined in Section 2.3.3 were utilized. The mean overlapping ratio for all scan registration pairs stood at 29.89%. Notably, despite the minimum overlapping ratio being 13.09%, the SSR was 100%, signifying the absence of registration failure cases.

Subsequently, the performance of the proposed approach was compared with two state-of-the-art multi-view registration approaches: MPCGR [85] and HL-MRF [28], using the acquired point cloud dataset. For fair comparisons, the point cloud dataset was filtered using a voxel grid with a 0.05 m spatial resolution and translated to the scan position to ensure the same conditions for initial translation approximation. Table 4 presents the performance for each method. Both approaches failed to register the multiple-point clouds, while the proposed approach succeeded. This is because conventional registration approaches do not account for cases of low overlap between point cloud pairs.

### 3.4. Comparison of Registration Results with Other Approaches

In this section, the performance of the proposed approach was compared with existing point cloud registration methods. Five modern point cloud registration methods were selected for this comparative analysis, including fast-descriptors-based [86], CoBigICP [87], 2D-line feature-based [88], LSG-CPD [89], and WES-ICP [90]. For fair comparisons, the input point cloud data were filtered using a voxel grid with a 0.05 m spatial resolution and translated to the scan position to maintain the consistent conditions for initial translation approximation. For the common parameter, the maximum iteration number for the registration process is set at 100. All the experiments are implemented in MATLAB 2023b on a single-thread system with an AMD Ryzen 7 3700X CPU @3.59 GHz processor and 16 GB of RAM.

The methods were tested on both the highest overlapped point cloud pair from the acquired point cloud dataset (Figure 14), which has a max overlapping ratio of 53.09%, and the lowest overlapped point cloud pair (Figure 15), which has a minimum overlapping ratio of 13.09%. The registration accuracy was evaluated using the RMSE value with the TPs, as in the previous section. The three evaluation metrics, SR (Equation (9)), RMSE (Equation (13)), and computation time, are used for the comparisons.

Table 5 presents the registration performance for each method on the highest overlapped point cloud pair. Among the compared methods, only three methods, including 2D-line feature-based [88], LSG-CPD [89], and the proposed approach, show the successful registration results with RMSE values under 0.1 m. The proposed approach represents the best registration accuracy and the fastest computation speed. Figure 16 illustrates the three successful registration results, along with color-coding of the reference point cloud (red) and the registered point cloud (green).

With those three successful approaches, the registration performance for each method on the lowest overlapped point cloud pair is compared. Table 6 presents the registration performance. It is shown that only the proposed approach presents a successful registration result with the fastest computation time. Figure 17 illustrates the three registration results, along with the color-coding of the reference point cloud (red) and the registered point cloud (green). This demonstrated that the proposed registration method is robust for the low-overlapping point cloud data.

### 3.5. Comparison of Registration Results on the Benchmark Datasets

In this section, the performance of the proposed approach was assessed using three datasets from different benchmarks: ETH–Office [69], RESSO—(e) and (i) [44]. Specifically, the ETH–Office dataset was captured from an indoor office room using a Faro Focus 3D laser scanner system. Given the confined nature of indoor spaces, larger overlaps are generally observed. RESSO—(e) and (i) were collected from indoor corridors using a Leica ScanStation C10. Each of these datasets provides the ground truth transformation parameters for every scan pose, allowing the transformation error to be determined using Equation (16) [28,43].
(16)$$\left\{\begin{array}{l} e^{r}=\cos^{-1}\left(\frac{tr\left(R^{g}{\left(R^{e}\right)}^{T}\right)-1}{2}\right)\\ e^{t}=\left\|t^{e}-t^{g}\right\| \end{array}\right.$$
The term is defined as a follow:
$R^{g},t^{g}:\;\mathrm{The}\;\mathrm{ground}\;\mathrm{truth}\;\mathrm{transformation}\;\mathrm{parameters}$$R^{e},t^{e}:\;\mathrm{The}\;\mathrm{estimated}\;\mathrm{transformation}\;\mathrm{parameters}\;\mathrm{from}\;\mathrm{the}\;\mathrm{registration}$$e^{r}:Rotation\;error$$e^{t}:Translation\;error$

As with Section 3.3, two state-of-the-art multi-view registration algorithms, MPCGR and HL-MRF, were selected to compare with the proposed method using the three benchmark datasets. The scan positions were assumed using the translation values of the ground truth, incorporating random errors up to ±0.05 m. Additionally, random angular errors along the *Z*-axis were artificially included, with values up to 5 degrees.

For the other registration approaches, the input point cloud was translated to the scan position to ensure consistent conditions for the initial translation approximation. Table 7 presents the comparison among these algorithms. The RMSE for rotation errors of the proposed approach is 0.102, 0.729, and 0.761 (deg), respectively, while the RMSE for translation errors is 0.038, 0.065, and 0.048 (m). The computation times for the three datasets are 33.21, 60.13, and 38.84 s, respectively. MPCGR failed for all datasets, while HL-MRF successfully registered the ETH–Office and RESSO—(i) datasets. However, the proposed method achieves state-of-the-art accuracy with a notable increase in time performance across all datasets. Figure 18 illustrates the three registration results.

## 4. Discussion

The proposed automated point cloud registration method for the stop-and-go system offers a reliable and expedited means of generating 3D point cloud maps of job sites, independent of human input. In contrast, manual registration methods not only require more time but also introduce uncertainties related to the registration quality, heavily dependent on the surveyor’s skills and experience.

The majority of man-made buildings typically adhere to the Manhattan world assumption, where walls are parallel or perpendicular to each other and perpendicular to ceilings and floors [91,92,93]. The proposed approach is specifically optimized for indoor environments that conform to this assumption, exhibiting enhanced registration performance compared to traditional methods. Given its design and efficacy, this approach is highly applicable to general indoor environments comprising multiple rooms and corridors.

Four distinct comparisons were conducted with conventional methods to validate different facets of the proposed approach: (1) To evaluate the overall registration performance using scan data acquired by the stop-and-go system; (2) To assess pairwise registration performance for point cloud pairs with the highest overlap; (3) To analyze pairwise registration performance for point cloud pairs with the lowest overlap; (4) To evaluate the overall registration performance with benchmark datasets. Each comparison involved several conventional methods, effectively showcasing the proposed approach’s efficiency across a variety of scenarios.

A common challenge in indoor environments, such as corridors, is the occurrence of low-overlap scan pairs, which often lead to registration failures. Traditional registration approaches struggle to overcome this issue. However, our proposed method demonstrates robust registration performance even with low-overlapped scan data, benefiting from the prior information provided by the mobile robot.

The validation carried out using the stop-and-go scanning system affirms the method’s efficacy in real-world indoor settings. This approach is novel in its combination of autonomous data acquisition with a quadruped walking robot and includes a comprehensive evaluation of the registration process. The study thus presents a valuable strategy for enhancing the efficiency of automated registration processes for such scanning systems.

The proposed method has some limitations. The first one is that this study heavily depends on the precondition of considering only rotation during the coarse registration, which is limited to around the *z*-axis. This assumption requires a reliable auto-leveling function in TLS. In fact, many commercial solutions for automated point cloud registration operate on the practical presumption that TLS can ensure automated leveling, subsequently performing their registration processes based on 2D registration, much like the method proposed in this study. Furthermore, it is crucial to note that advancements in TLS technology are steadily improving its automated leveling capabilities. To counteract potential problems stemming from mis-leveling, a preprocessing step for fine *Z*-axis alignment has been additionally integrated into this study, despite the existing limitations of current auto-leveling technology.

The second is that the proposed approach is exclusively applicable to man-made indoor environments adhering to the Manhattan world condition. While the majority of indoor building environments continue to conform to the Manhattan world condition, modern architecture is increasingly incorporating nonlinear elements, such as curved walls and sliding structural components. Addressing and including these types of nonlinear structural elements in the approach would contribute to enhanced performance and wider applicability.

## 5. Conclusions

In this study, an automated point cloud registration approach optimized for a stop-and-go scanning system is proposed to efficiently register the 3D scan data using the localization of the scanning system as prior knowledge. The proposed approach consists of three primary stages. In the initial phase, the input point cloud is obtained from the stop-and-go scanning system in the previous research [16], and perpendicular constrained wall planes are extracted. This involves voxel filtering and fine alignment to the *Z*-axis, followed by the identification of wall-related points based on their normal vectors. The subsequent step employs the M-SAC algorithm to extract wall-plane models with perpendicular constraints. During the second phase, the coarse registration step involves aligning wall planes constrained perpendicularly, and the initial position of the scan system is utilized. To ascertain potential rotation angles, the alignment of normal vectors on each wall plane is assessed, and the calculation of overlapping scores for these angles occurs through an analysis of point-to-point displacement. The third phase encompasses executing fine registration using the proposed horizontality constrained ICP algorithm. The matching patches extracted for the overlapping score in the second phase are utilized for the fine registration process.

The major contributions of this study in the field of automatic point cloud registration are as follows: Firstly, the proposed approach exhibits robust registration results in scenarios with low-overlapped scan data. Secondly, it demonstrates superior registration performance compared to conventional methods. Thirdly, validation using the stop-and-go scanning system confirms the reliability of the proposed approach for scan registration in real-world indoor environments. The proposed approach is the first to tackle automatic registration with autonomous data acquisition using a quadruped walking robot, along with a comprehensive evaluation of the registration process. Lastly, to the best of our knowledge, no studies have yet focused on point cloud registration with an acquired point cloud dataset from stop-and-go scanning systems. As such, this study is valuable in proposing a strategy to enhance the efficiency of the automated registration process for these scanning systems.

Future work will address the limitations of the proposed approach. This method is tailored for registering point clouds from scenes predominantly featuring planar structures, thus making it especially suitable for scans of man-made indoor environments. Therefore, the approach might struggle with outdoor scans or scans of individual objects characterized by curved surfaces. Additionally, precise and accurate localization performance of the scanning system is a prerequisite, which can be facilitated by state-of-the-art mobile platforms.

## Figures and Tables

**Figure 1 sensors-24-00138-f001:**
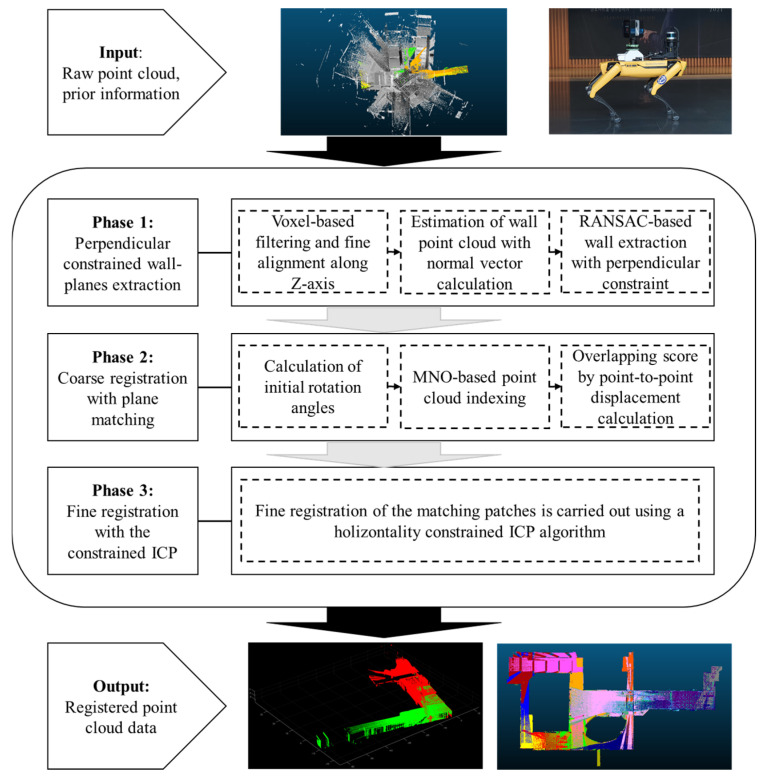
The workflow for the proposed point cloud registration approach.

**Figure 2 sensors-24-00138-f002:**
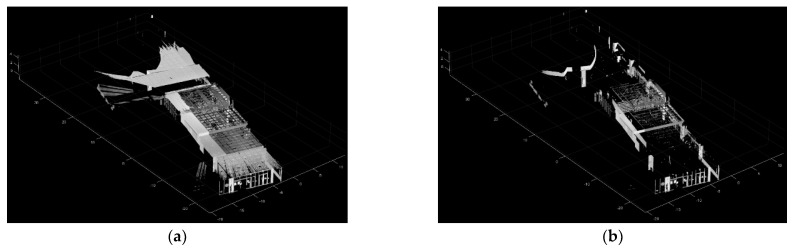
An example of the estimated wall point cloud. (**a**) Input point cloud; (**b**) The estimated wall point cloud. Points belonging to wall structures have been segmented, along with some noise points that are not part of the walls.

**Figure 3 sensors-24-00138-f003:**
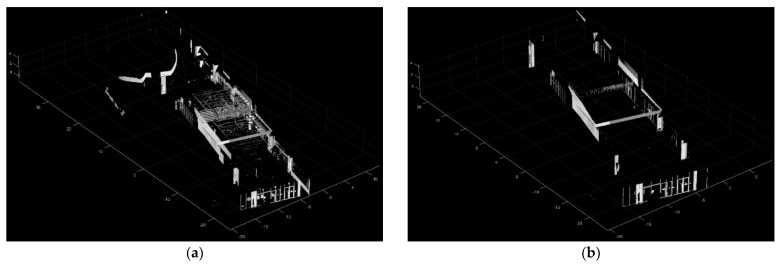
An example of the M-SAC-based wall extraction. (**a**) Input point cloud; (**b**) The extracted wall-point cloud. Non-wall points distinctly removed.

**Figure 4 sensors-24-00138-f004:**
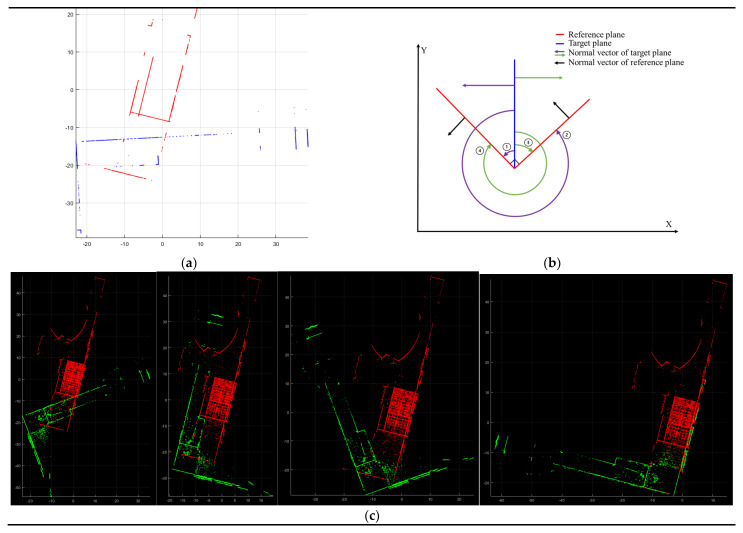
A set of rotation candidates for the target plane matching in top view. (**a**) The reference wall planes in red and the target wall planes in blue; (**b**) A set of rotation candidates: four rotation angles around the *Z*-axis; (**c**) Four temporary transformed target point cloud (green) cases generated based on the set of rotation candidates.

**Figure 5 sensors-24-00138-f005:**
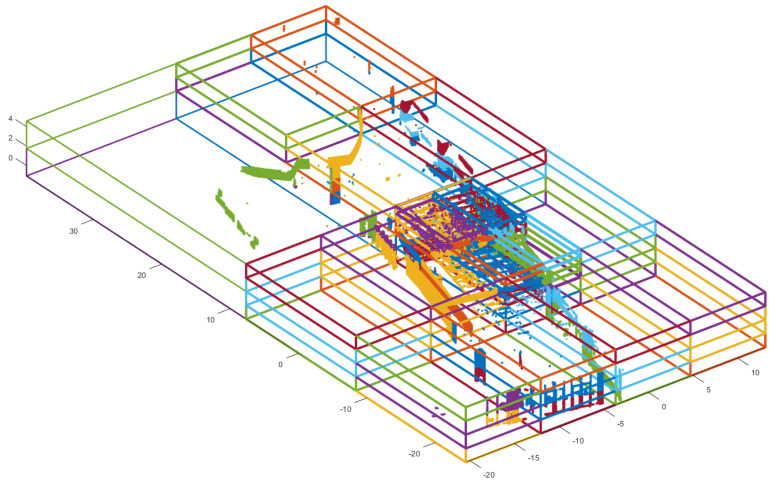
MNO indexing result with the $w_{P}$ of the reference point cloud. Outcome of point cloud indexing achieved through utilization of the MNO structure. Points indexed within the same MNO node are depicted in the same color.

**Figure 6 sensors-24-00138-f006:**
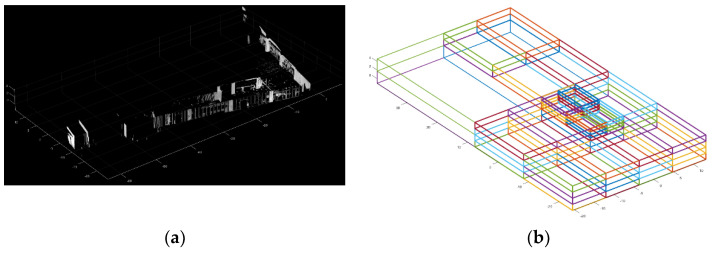
An example of the indexing process with the constructed MNO structure. (**a**) Input target point cloud; (**b**) The constructed MNO structure; (**c**) The indexed target point cloud. The points within the same MNO node are depicted in the same color, while the points out of the MNO structure are depicted in black.

**Figure 7 sensors-24-00138-f007:**
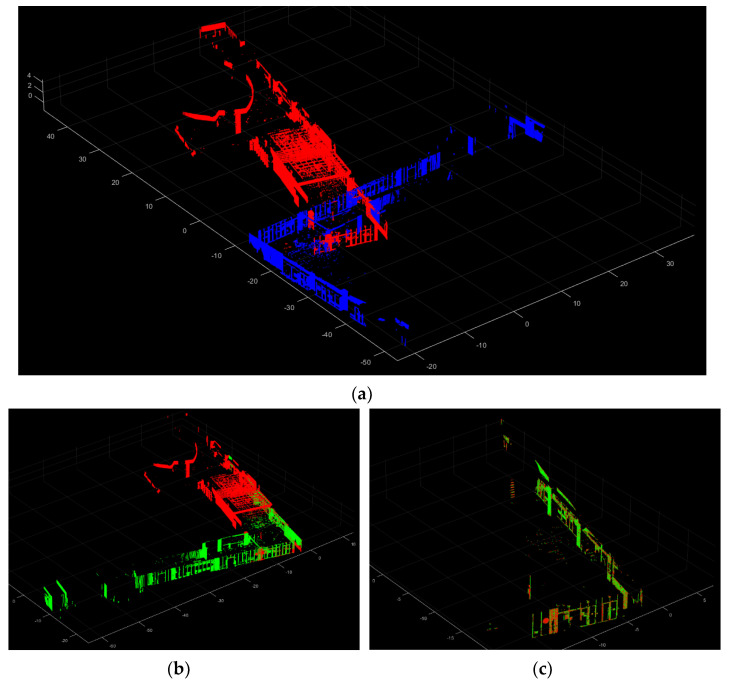
An example of the coarse registration result with the overlapping score. An example of the coarse registration process with a displacement threshold of 0.5 m. (**a**) The reference point cloud (red) and the target point cloud (blue); (**b**,**d**,**f**,**h**) The results (green) after applying each potential rotation angle; (**c**,**e**,**g**,**i**) The overlapping points between the reference point cloud (red) and the transformed target point cloud (green) pairs; (**j**) The coarse registration outcome. In this case, the overlapping points (**c**) are identified as the matching patches for the subsequent fine registration.

**Figure 8 sensors-24-00138-f008:**
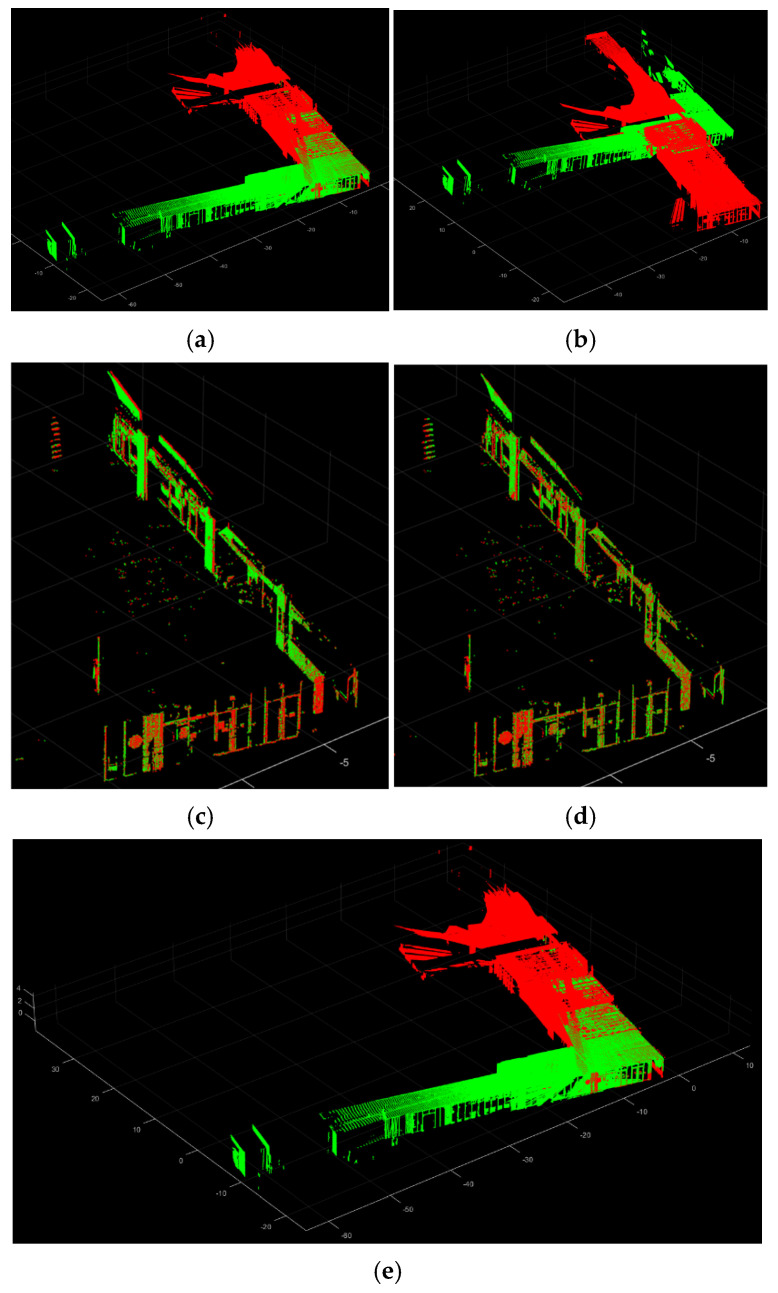
An example of the fine registration. (**a**) Raw point cloud; (**b**) Failed registration result; (**c**) The input matching patches; (**d**) the fine registration of the matching patches; (**e**) The fine registration result.

**Figure 9 sensors-24-00138-f009:**
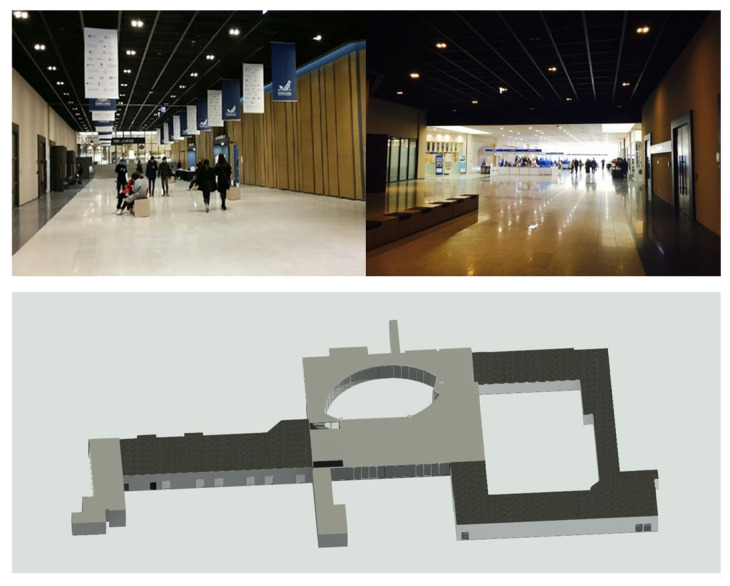
The scenery and BIM of the test site.

**Figure 10 sensors-24-00138-f010:**
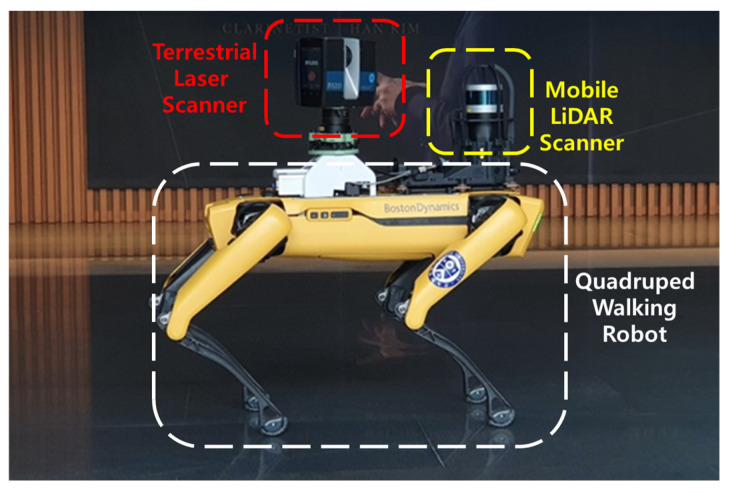
The stop-and-go scanning system developed in [16]. The quadruped walking robot with MLS for localization of the robot and TLS for the stop-and-go scanning process.

**Figure 11 sensors-24-00138-f011:**
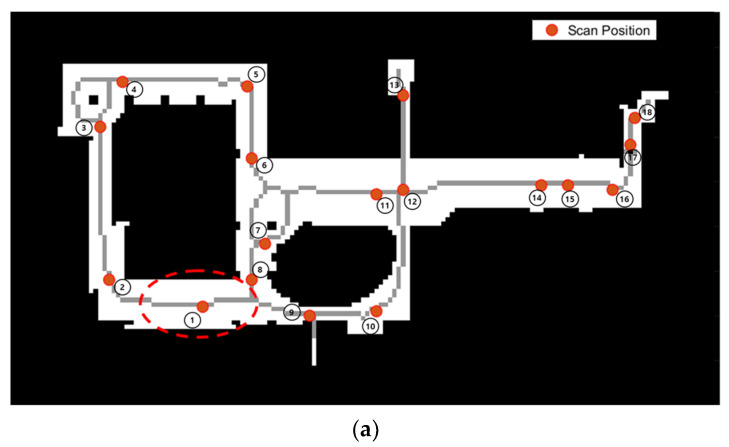
Control points for the test site. (**a**) Distribution map of scanning positions and CPs. The number of scan positions depicts the registration order. (**b**) View of CPs within blue dotted circles, captured by the total station; (**c**) View of CPs points within blue dotted circles, extracted from the reference point cloud data.

**Figure 12 sensors-24-00138-f012:**
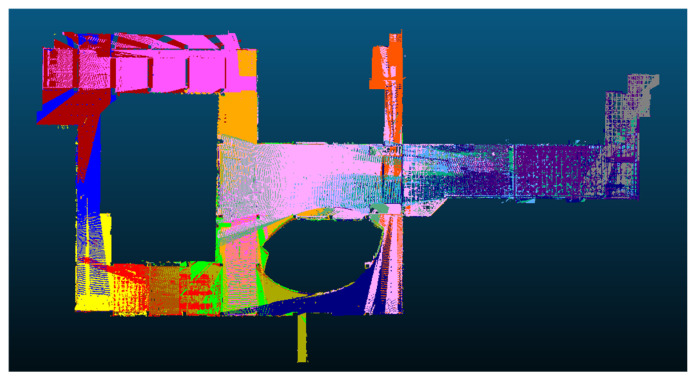
The visualization of registered point cloud. The point clouds are colored according to the difference in each scan station.

**Figure 13 sensors-24-00138-f013:**
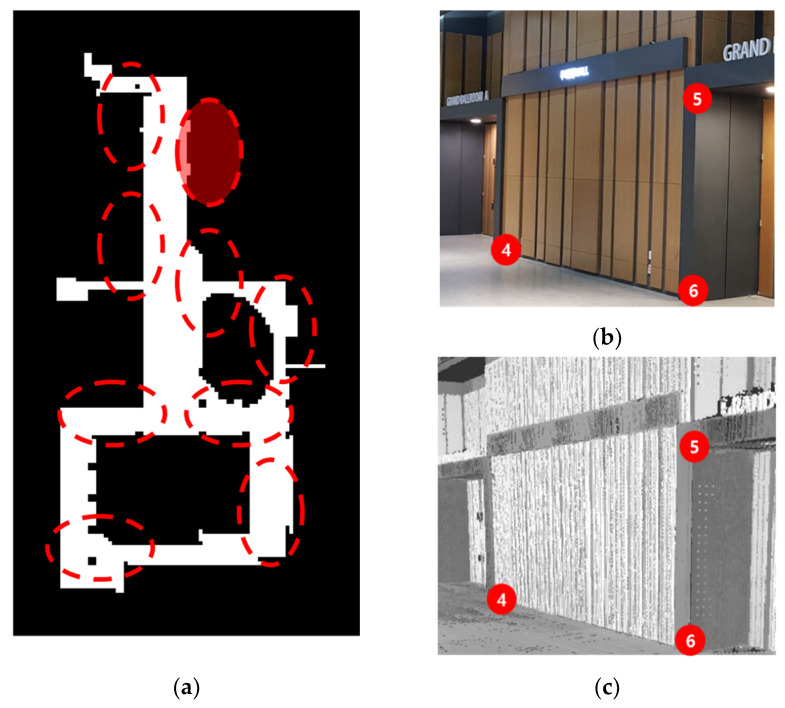
Assessment of target points. (**a**) Distribution map of target points. Target points are indicated by red dotted circles; (**b**) View of target points within a red dotted circle, captured by the total station; (**c**) View of target points within a red dotted circle, extracted from the registered point cloud data.

**Figure 14 sensors-24-00138-f014:**
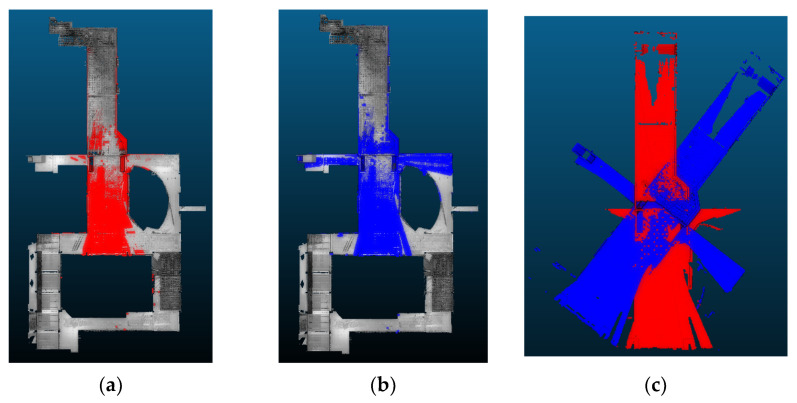
The highest overlapped scan pair. (**a**) The reference point cloud (red) in the entire point cloud scene; (**b**) The target point cloud (blue) in the entire point cloud scene; (**c**) The raw target point cloud with the reference point cloud.

**Figure 15 sensors-24-00138-f015:**
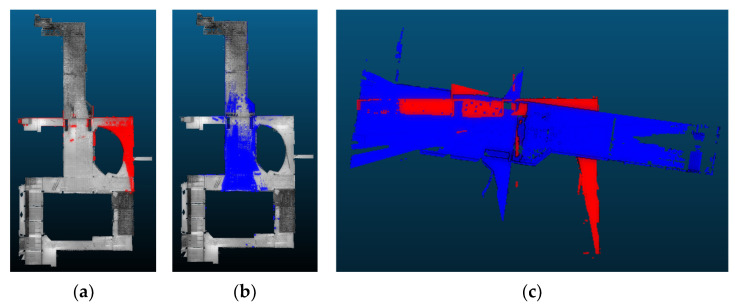
The lowest overlapped scan pair. (**a**) The reference point cloud (red) in the entire point cloud scene; (**b**) The target point cloud (blue) in the entire point cloud scene; (**c**) The raw target point cloud with the reference point cloud.

**Figure 16 sensors-24-00138-f016:**
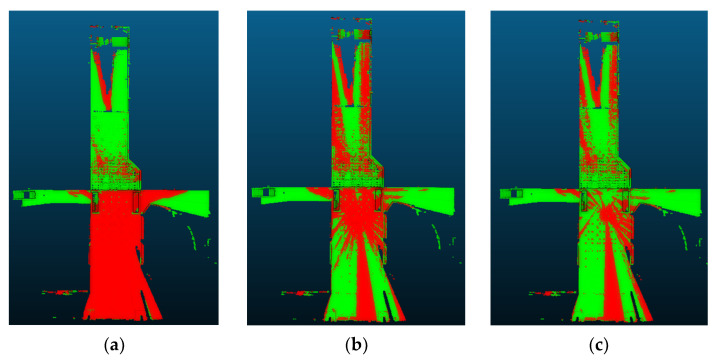
The registration results for the highest overlapped scan pair. (**a**) 2D-line-featured-based approach [88]; (**b**) LSG-CPD [89]; (**c**) The proposed approach.

**Figure 17 sensors-24-00138-f017:**
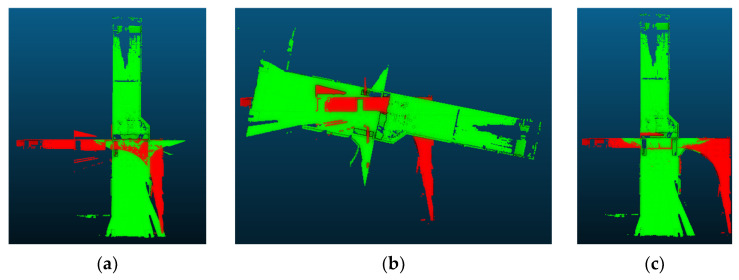
The registration results for the lowest overlapped scan pair. (**a**) 2D-line-featured-based approach [88]; (**b**) LSG-CPD [89]; (**c**) The proposed approach.

**Figure 18 sensors-24-00138-f018:**
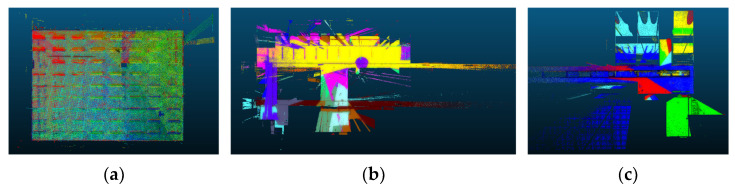
The registration results of the proposed approach on the bench datasets. (**a**) ETH–Office [69]; (**b**) RESSO—(e) [44]; (**c**) RESSO—(i) [44].

**Table 1 sensors-24-00138-t001:** Summary of the literature review on the stop-and-go scanning system in terms of its registration method.

	Data Acquisition			Prior Information	Registration	
	Platform	Overlap	Autonomous	Data	Approach	Evaluation
Blaer and Allen 2007 [20]	UGV	-	O	-	Target-and-manual-based	X
Kurazume et al. 2017 [58]	Multi-UGVs	-	O	Localization data	Target-and-manual-based	X
Liu et al. 2020 [59]	Vehicle	-	X	-	Manual	X
Park et al. 2023 [16]	Quadruped robot	-	O	-	Manual	X
Lin et al. 2013 [63]	MMS	-	-	IMU, GPS	ICP	X
Mohammed et al. 2014 [64]	-	High (>50%)	-	RGB image	SURF, ICP	O
Ge et al. 2019 [61]	-	High (>50%)	-	Panorama image	Visual SLAM	O
Knechtel et al. 2022 [65]	-	High (>50%)	-	Scan planning	ICP	X
Chow et al. 2014 [21]	Handcart	High (>50%)	X	IMU, RGB-D	SLAM, ICP	X
Borrmann et al. 2014 [67]	UGV	High (>50%)	O	Odometry	SLAM	X
Zhong et al. 2016 [60]	Vehicle	High (>50%)	X	IMU, GNSS	System Calibration	X
Prieto et al. 2017 [23]	UGV	High (>50%)	O	Localization data	ICP	X
Proposed method	Quadruped robot	Low (<30%)	O	Localization data	Plane-based, ICP	O

**Table 2 sensors-24-00138-t002:** The parameters of the scan system.

Parameter	Value
Scanner height	1 m
Lase range	20 m
Vertical field of view	−150°–+150°
Horizontal field of view	−180°–+180°

**Table 3 sensors-24-00138-t003:** The description of the registration performance.

Description	Value
# of Scans	18
# of Points	186 million
Computation Time (s)	424.2
Max Overlapping Ratio (%)	53.09
Min Overlapping Ratio (%)	13.09
Average Overlapping Ratio (%)	29.89
SSR (%)	100
Positional Accuracy (RMSE, m)	0.044

**Table 4 sensors-24-00138-t004:** The comparison of the registration performance.

Method	SRR (%)	Positional Accuracy (m)	Time (s)
MPCGR [85]	0	-	-
HL-MRF [28]	0	-	-
Proposed	100	0.044	424.2

**Table 5 sensors-24-00138-t005:** Comparison of registration performances with the highest overlapped scan pair.

Method	SRR(O/X)	Positional Accuracy (m)	Time (s)
Fast-descriptors-based [86]	X	0.137	561.53
CoBigICP [87]	X	7.556	1257.91
2D-line feature-based [88]	O	0.061	826.36
LSG-CPD [89]	O	0.079	982.03
WES-ICP [90]	X	5.764	1572.15
Proposed	O	0.030	53.95

**Table 6 sensors-24-00138-t006:** Comparison of registration performances with the lowest overlapped scan pair.

Method	SRR(O/X)	Positional Accuracy (m)	Time (s)
2D-line feature-based [88]	X	16.453	820.50
LSG-CPD [89]	X	17.817	283.33
Proposed	O	0.029	29.60

**Table 7 sensors-24-00138-t007:** Comparison of registration performances on the benchmark datasets.

Dataset	Method	SRR(O/X)	Rotation Error (degree)	Translation Error (m)	Time (s)
ETH–Office [69]	MPCGR [85]	X	-	-	-
	HL-MRF [28]	O	0.648	0.041	233.28
	Proposed	O	0.102	0.038	33.21
RESSO—(e) [44]	MPCGR [85]	X	-	-	-
	HL-MRF [28]	X	-	-	-
	Proposed	O	0.729	0.065	60.13
RESSO—(i) [44]	MPCGR [85]	X	-	-	-
	HL-MRF [28]	O	1.316	0.075	923.19
	Proposed	O	0.761	0.048	38.84

## Data Availability

The data that support the findings of this study are available from the corresponding author, upon reasonable request.

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
