# Peer review of "Automated Point Cloud Registration Approach Optimized for a Stop-and-Go Scanning System"

_sensors, 2023, doi:10.3390/s24010138_

Round 1
Reviewer 1 Report
Comments and Suggestions for Authors
This manuscript presents an automated point cloud registration approach optimized for a Stop-and-Go scanning system based on a quadruped walking robot. Experimental results show that the proposed method surpasses traditional techniques, however, the following parts can be improved.
1. What is the impact and criterion of the set number and location of control points and scan positions?
2. Figure 1. shows the workflow for the proposed point cloud registration approach. However, which part is the main innovation of your manuscript? Or just a combination of traditional methods?
3. It says that the proposed method is an automated method. However, the workflow of the proposed approach involves multiple processes. Does any process rely on manual intervention? Can the proposed method be automated in all of the processes?
4. The proposed method is applicable to specific indoor environment conditions, and the assumption is based on the walls. However, many scenarios don’t satisfy the assumption. What are the conditions and reasons for the specific application of the proposed method? What are the disadvantages and limitations of the proposed approach? Separate discussions can be added.
5. Most simulations are visually qualitative evaluations. Why are the comparison methods in Table 3、Table 4、Table 5 and Table 6 different?
Comments on the Quality of English LanguageMinor editing of English language.
Author Response
We would like to thank the editor and the reviewers for their valuable comments, which have helped us improve the manuscript. Below, please find our item-by-item responses to the reviewers’ comments.
REVIEWER #1: This manuscript presents an automated point cloud registration approach optimized for a Stop-and-Go scanning system based on a quadruped walking robot. Experimental results show that the proposed method surpasses traditional techniques, however, the following parts can be improved.
Response: We thank you for your valuable comments and suggestions. Also, we would like to note that the manuscript was revised according to your comments.
COMMENT #1: What is the impact and criterion of the set number and location of control points and scan positions?
Response: The number of the control points (CP) affects the accuracy of transforming the reference point cloud from its local coordinate system to the global coordinate system of the scanning system. Specifically, as the number of CPs increases, the accuracy and precision of the coordinate transformation improve, adhering to the principles of least squares adjustment. For 3D coordinate transformation, a minimum of three control points (CPs) are mandatory. In this study, seven CPs were surveyed and employed for least squares adjustment to enhance the transformation performance. The locations of these CPs were strategically chosen to be easily identifiable by the total station, such as the corners of columns, doors, and walls.
Regarding the number and location of scan positions, these were determined based on our previous research on scan planning (Park et al. 2023). The optimal scan positions were calculated through a 3D visibility analysis involving ray tracing of the BIM geometry, and the number of scan positions was selected using the proposed optimization algorithm.
We added relevant explanations to Section 3.2 (Line #591-596) and Section 3.3 (Line #607-611).
Park, S., Yoon, S., Ju, S., & Heo, J. (2023). BIM-based Scan Planning for Scanning with a Quadruped Walking Robot. Automation in Construction, vol.152, pp.104911
COMMENT #2: Figure 1. shows the workflow for the proposed point cloud registration approach. However, which part is the main innovation of your manuscript? Or just a combination of traditional methods?
Response: The proposed point cloud registration approach presents originality and distinctiveness from traditional methods, particularly in Phase 2. In this phase, feature points for registration are determined by calculating point-to-point displacement for the entire points of a point cloud pair. This is achieved through Modifiable Nested Octree (MNO) based point cloud indexing. Traditional approaches like ICP-based, typically employ KDtree-based KNN for calculating point pair distances, but this approach often faces reduced computational efficiency with increasing point cloud sizes. The MNO-based approach, on the other hand, enables fast and comprehensive point-to-point displacement calculation, allowing for robust matching point generation by comparing every point in each point cloud. This phase is a key factor contributing to the fast and robust registration performance of the proposed approach. The experimental results further substantiate this, demonstrating that the proposed method exhibits enhanced computational performance compared to conventional approaches.
COMMENT #3: It says that the proposed method is an automated method. However, the workflow of the proposed approach involves multiple processes. Does any process rely on manual intervention? Can the proposed method be automated in all of the processes?
Response: Yes, once the data is input, the proposed method does not rely on any manual intervention. The proposed method can be automated in all of the processes in the workflow. We added relevant explanations to Section 2.1 (Line #259-260)
COMMENT #4: The proposed method is applicable to specific indoor environment conditions, and the assumption is based on the walls. However, many scenarios don’t satisfy the assumption. What are the conditions and reasons for the specific application of the proposed method? What are the disadvantages and limitations of the proposed approach? Separate discussions can be added.
Response: Thank you for your valuable comment. The majority of man-made buildings typically adhere to the Manhattan world assumption, where walls are parallel or perpendicular to each other, and perpendicular to ceilings and floors. The proposed approach is specifically optimized for indoor environments that conform to this assumption, exhibiting enhanced registration performance compared to traditional methods. Given its design and efficacy, this approach is highly applicable to general indoor environments comprising multiple rooms and corridors. We added the relevant explanations to discussion section as your suggestion (Lines #749-755).
The disadvantages and limitations of the proposed approach regarding the Manhattan world assumption is detailed in discussion section (Lines #786-792). Also, the future work for this limitation is mentioned in conclusion section (Lines #822-826).
COMMENT #5: 5. Most simulations are visually qualitative evaluations. Why are the comparison methods in Table 3、Table 4、Table 5 and Table 6 different?
Response: In this study, four distinct comparisons were conducted with conventional approaches to validate various aspects of the proposed method.
The first comparison (Table 4) involved the proposed approach and two state-of-the-art multi-view registration methods, designed for registering multiple point clouds from each scan station. The two methods failed to register multiple point clouds, while the proposed method succeeded.
Secondly, the proposed method was compared with five existing pairwise registration methods (Table 5), using the point cloud pair with the highest overlap from the dataset acquired by the Stop-and-Go scanning system. Among these, only three methods achieved Successful Registration (SR), and the proposed method stood out in terms of both RMSE and processing time.
The third comparison (Table 6) specifically focused on evaluating the registration performance for the point cloud pair with the lowest overlap. While it would have been possible to use the same five methods as in the second comparison, it was anticipated that methods which failed in the highest overlap pair would also fail in the lowest overlap pair. Therefore, the choice was made to only include the two methods that had succeeded in the previous comparison. The results from this comparison demonstrated that only the proposed method was able to achieve a successful registration outcome, and it did so with the most efficient computation time.
The final comparison (Table 7) was benchmarked against three different datasets. For these comparisons, using the same two multi-view registration methods from the first comparison. The proposed method showed state-of-the-art accuracy and significantly faster processing speed across all datasets.
Each comparison with several conventional methods was tailored to test specific capabilities of the proposed method, demonstrating its efficiency in various scenarios.
We added relevant explanations to Section 4 (Line #756-763).

Reviewer 2 Report
Comments and Suggestions for Authors
The submited work presents a methodology for automated point cloud registration. The alignment of multiple point clouds from local coordinate systems into one unified coordinate system involves determining three 3D rotation parameters and 3 translation values, for each local point cloud.
The proposed approach solve the special case where there is only one rotation (z-axis) and a translation are needed, which is suitable to align point clouds from mobile LiDAR scanning of indoor environments.
In my opinion, the approach is not innovative, however authors claim that the results from the proposed method compares better than other known methods, which is itself an improvement for the state of the art of 3D modelling.
The presentation of the methodology and the results are sound.
Author Response
We would like to thank the editor and the reviewers for their valuable comments, which have helped us improve the manuscript. Below, please find our item-by-item responses to the reviewers’ comments.
REVIEWER #2: The submited work presents a methodology for automated point cloud registration. The alignment of multiple point clouds from local coordinate systems into one unified coordinate system involves determining three 3D rotation parameters and 3 translation values, for each local point cloud.
The proposed approach solve the special case where there is only one rotation (z-axis) and a translation are needed, which is suitable to align point clouds from mobile LiDAR scanning of indoor environments.
In my opinion, the approach is not innovative, however authors claim that the results from the proposed method compares better than other known methods, which is itself an improvement for the state of the art of 3D modelling.
The presentation of the methodology and the results are sound.
Response: We would like to thank for your encouragement.

Reviewer 3 Report
Comments and Suggestions for Authors
Abstract should contain more information from the field of results, concrete values, accuracy, etc., so as to convince to delve into the content of the publication - without engineering arguments, i.e. values, it is hardly convincing
line 9 and key words - rather Terrestrial Laser Scanning than Terrestrial LiDAR scanning, the line 43 also - please correct whole manuscript
line 43 - the same with MLS
line 49/50 - as an example of differences in data density and accuracy you can use: https://doi.org/10.5194/isprs-archives-XLVIII-1-W3-2023-205-2023
line 95 - MMS - acronym to be developed
you can also mention about FAMFR algorithm https://doi.org/10.1186/s40494-023-01018-y
line 147 - there should be GNSS instead of GPS, I think
lines 180--188 - please do not quote in blocks , 410, 412 the same
table 1 - it would be good if the descriptions "high" and "low" could be supported by some numerical values, if not by one digit then at least by ranges
fig.4 - mentioned targer wall-planes in blue are completely invisible - please correct
it will be good to add some description about: avg distance between scan positions or length of whole path of the scanning system, what parameters in scanners were set up? what density of the point clouds were produced
Author Response
We would like to thank the editor and the reviewers for their valuable comments, which have helped us improve the manuscript. Below, please find our item-by-item responses to the reviewers’ comments.
REVIEWER #3
COMMENT #1: Abstract should contain more information from the field of results, concrete values, accuracy, etc., so as to convince to delve into the content of the publication - without engineering arguments, i.e. values, it is hardly convincing.
Response: Thank you for your valuable comment. We added more information from the experimental results like accuracy, registration rate and computational performance as your suggestion (Lines #17-20).
COMMENT #2: line 9 and key words - rather Terrestrial Laser Scanning than Terrestrial LiDAR scanning, the line 43 also - please correct whole manuscript
Response: As your suggestion, we corrected the term, Terrestrial LiDAR Scanning, to Terrestrial Laser Scanning through whole manuscript (Lines #10, 23 and 47).
COMMENT #3: line 43 - the same with MLS
Response: We corrected the term, Mobile LiDAR Scanning, to Mobile Laser Scanning (Lines #47).
COMMENT #4: line 49/50 - as an example of differences in data density and accuracy you can use: https://doi.org/10.5194/isprs-archives-XLVIII-1-W3-2023-205-2023
Response: Thank you for introducing the reference paper. We reviewed the papers that you suggested and have cited this paper to support the assertion in Line # 54-55.
COMMENT #5: line 95 - MMS - acronym to be developed
Response: We add the term Mobile Mapping System to explain the acronym MMS (Lines #99 and 154).
COMMENT #6: line 172. Define the concept visibility (in 2D or in 3D?)
Response: In the work of Prieto et al., the visibility analysis conducted for their Next Best Scan (NBS) algorithm is based on a 2D vertical image projection. This approach involves vertically projecting the point cloud onto each 3D model of a wall component, using a resolution of 5 cm/pixel. The visibility is then calculated from the number of pixel points and the overall size of the image. We added relevant explanations to Line #181-185.
COMMENT #7: you can also mention about FAMFR algorithm https://doi.org/10.1186/s40494-023-01018-y
Response: Thank you for your valuable comment. We reviewed the papers that you suggested and have added the contents and findings of these papers to the manuscript as a related work (Lines #167-171).
COMMENT #8: line 147 - there should be GNSS instead of GPS, I think
Response: We corrected the term, GPS, to Global Navigation Satellite System (GNSS) (Lines #152-153).
COMMENT #9: lines 180--188 - please do not quote in blocks , 410, 412 the same
Response: We removed citations in blocks as your suggestion (Lines #193-202) (Lines #422 and 424).
COMMENT #10: table 1 - it would be good if the descriptions "high" and "low" could be supported by some numerical values, if not by one digit then at least by ranges
Response: Thank you for your comment. We add the ranges for supporting the description “high” as > 50%, “low” as < 30%.
COMMENT #11: fig.4 - mentioned target wall-planes in blue are completely invisible - please correct
Response: In Figure 4(a), target wall-plane in blue is visible. Figure 4(c) present the coarse registration results, so it illustrates the registered target wall-plane in green color and the reference wall-plane in red color. We changed the background color of Figure 4(a) to clearly figure out the target wall-planes.
COMMENT #12: it will be good to add some description about: avg distance between scan positions or length of whole path of the scanning system, what parameters in scanners were set up? what density of the point clouds were produced
Response: Thank you for your comment. The scanning system in this study follows the Stop-and-Go system which was developed at our previous work (Park et al. 2023). The below table depicts parameter value of Faro scanner using in this study. The average distance between scan positions is 17.68 m. In this study, we adopted voxel-based filtering approach, so density of the point cloud is 5 cm. We added relevant explanations and Table 2 to Section 3.2 (Line #591-596 and 600).
Table 2. The parameters of the scan system
|
Parameter |
Value |
|
Scanner height |
1 m |
|
Laser range |
20 m |
|
Vertical field of view (Faro Focus 3D) |
-150° - +150° |
|
Horizontal field of view (Faro Focus 3D) |
-180° - +180° |

Round 2
Reviewer 1 Report
Comments and Suggestions for Authors
The manuscript can be accepted in present form.
Comments on the Quality of English LanguageMinor editing of English language.